# Fungal Biomarkers in Traditional Starter Determine the Chemical Characteristics of Turbid Rice Wine from the Rim of the Sichuan Basin, China

**DOI:** 10.3390/foods12030585

**Published:** 2023-01-30

**Authors:** Lanchai Chen, Wenliang Xiang, Xuemei Liang, Junyu Liu, Haoyu Zhu, Ting Cai, Qing Zhang, Jie Tang

**Affiliations:** 1School of Food and Bioengineering, Xihua University, Chengdu 610039, China; 2Key Laboratory of Food Microbiology of Sichuan, Xihua University, Chengdu 610039, China; 3Chongqing Key Laboratory of Speciality Food Co-Built by Sichuan and Chongqing, Xihua University, Chengdu 610039, China; 4Sichuan Vocational School of Commerce, Chengdu 611731, China

**Keywords:** Qu, Sichuan basin, turbid rice wine, fungal biomarker, volatile compound, correlation analysis

## Abstract

The fungal community in Qu plays a key role in the formation of turbid rice wine (TRW) style. The Sichuan Basin and its surrounding areas have become one of the main TRW production regions in China; however, the fungal community in Qu and how they affect the characteristics of TRW remain unknown. Therefore, this study provided insight into the fungal biomarkers in Qu from Guang’an (GQ), Dazhou (DQ), Aba (AQ), and Liangshan (LQ), as well as their relationships with compounds in TRW. The main biomarkers in GQ were *Rhizopus arrhizus*, *Candida glabrata*, *Rhizomucor pusillus*, *Thermomyces lanuginosus* and *Wallemia sebi*. However, they changed to *Saccharomycopsis fibuligera* and *Mucor indicus* in DQ, *Lichtheimia ramose* in AQ, and *Rhizopus microsporus* and *Saccharomyces cerevisiae* in LQ. As a response to fungal biomarkers, the reducing sugar, ethanol, organic acids, and volatile compounds were also changed markedly in TRWs. Among important volatile compounds (VIP > 1.00), phenethyl alcohol (14.1–29.4%) was dominant in TRWs. Meanwhile, 3-methyl-1-butanol (20.6–56.5%) was dominant in all TRWs except that fermented by GQ (GW). Acetic acid (29.4%) and ethyl palmitate (10.1%) were dominant in GW and LW, respectively. Moreover, GQ biomarkers were positively correlated with acetic acid and all unique important volatile compounds in GW. DQ biomarkers had positive correlations with unique compounds of acetoin and ethyl 5-chloro-1,3,4-thiadiazole-2-carboxylate in DW. Meanwhile, the AQ biomarkers were positively correlated with all AW unique, important, and volatile compounds. Although there were not any unique volatile compounds in LW, 16 important volatile compounds in LW were positively related to LQ biomarkers. Obviously, biomarkers in different geographic Qu played vital roles in the formation of important volatile compounds, which could contribute specific flavor to TRWs. This study provided a scientific understanding for future efforts to promote the excellent characteristics of TRW by regulating beneficial fungal communities.

## 1. Introduction

Turbid rice wine (TRW) is one of the most popular alcoholic beverages in east Asia [1]. In China, it is consumed mainly by Han and ethnic minorities in the southern rice-growing regions because of its rich nutrition, such as amino acids, polypeptides, vitamins, and other bioactive components [2,3]. TRW is generally made by semisolid-state fermentation of glutinous rice, millet or other cereals [4] with TRW starter (Qu) in a relatively closed pottery jar [5]. In this case, the stand or fall of Qu directly affects the quality of TRW products. However, Qu is a common spontaneous fermentation product and its microorganism structures are affected by various factors, such as climate, temperature, humidity and natural environment microorganisms, etc. [6,7]. Thus, the community structure of Qu usually has its own unique characteristics compared with different regions, which is probably the main reason that determines the style of TRW in different regions. In recent decades, numerous studies have shown that fungi in Qu are more important than bacteria during the TRW fermentation, and fungal community structure is closely related to TRW quality characteristics [6,8]. However, it is not clear which fungi mainly respond to this correlation.

In China, TRW fermentation has a history of about 4000–5000 years, and it was first introduced from the Central Plains to the Han inhabited areas of the parallel fold ridge valley belt in the eastern Sichuan Basin, and then spread to the entire Han area in the Sichuan Basin during the Han Dynasty (250–100 BC). Later, with the civil war and cultural exchanges, it further spread to ethnic minority areas around the Sichuan Basin, to the Yi inhabited areas in the northern Yunnan–Kweichow Plateau, located south of the Sichuan Basin, during the Three Kingdoms period (220–280 AD); and to the areas inhabited by the Gyalrong Tibetans in the eastern Qinghai–Tibet Plateau, located west of the Sichuan Basin, during the Ting Dynasty (636–1911 AD). At present, the Sichuan Basin and its surrounding areas have developed into one of the main production areas of Chinese TRW. In these regions, due to the differences in geographical characteristics, climate conditions, temperature, humidity, and other environmental conditions, the different TRW brewing regions, although firstly originated from the parallel fold ridge valley belt in the eastern Sichuan Basin, have developed their own unique Qu, which leads to the distinct regional characteristics of TRW from these areas. The Han’s TRW is mainly suitable for the Han people of China, Yi’s TRW is suitable for the people of Yi and other ethnic minorities in the northern Yunnan–Kweichow Plateau, and Tibetan’s TRW is suitable for the people of Gyalrong, Kham Tibetan, and the Qiang ethnic minority in the eastern Qinghai–Tibet Plateau. However, it is unclear which fungal structures characterize the uniqueness of these Qu and how they respond to the regional features of their TRW.

Therefore, this study focused on how the fungal biomarkers of Qu respond to the regional characteristics of TRW in the rim of the Sichuan Basin, China. The fungal structures were explored in Qu from the parallel fold ridge valley belt, the northern Yunnan-Kweichow Plateau, and the eastern Qinghai–Tibet Plateau around the rim of the Sichuan Basin, respectively. In addition, the reducing sugar, ethanol, organic acids, and volatile compounds were further analyzed in TRW fermented by these Qu. To date, this is the first systematic report on the correlation between fungal biomarkers in Qu and the style of corresponding TRW from the rim of the Sichuan Basin in Southwest China.

## 2. Materials and Methods

### 2.1. Sample Collection

Four samples of starters (Qu) were obtained from different geographical areas around the Sichuan Basin in China (Figure 1), including Guang’an City (GQ, local market) and Dazhou City (DQ, Sichuan Dongliu Rice Wine Co., Ltd., Chengdu, China) in the parallel fold ridge valley belt, Aba Tibetan and Qiang Autonomous Prefecture (AQ, Sichuan Aba Heishui Wines Co., Ltd., Chengdu, China) in the eastern Qinghai–Tibet Plateau and Liangshan Yi Autonomous Prefecture (LQ, Sichuan Liangshan Dezhou Wines Co., Ltd., Chengdu, China) in the northern Yunnan–Kweichow Plateau. All samples were stored at 4 °C before analysis.

### 2.2. DNA Extraction and Sequencing

Total genomic DNA of Qu was extracted using PowerSoil^®^ DNA isolation kit (Mobio, Mobio Technologies Inc., Vancouver, BC, Canada) and analyzed by agarose gel electrophoresis and NanoDrop 2000c spectrophotometer (Thermo Scientific, Thermo Fisher Scientific Inc., New York, NY, USA). The V4 hypervariable regions of fungal 18S rRNA gene were amplified by the 528F (5′-GCGGTAATTCCAGCTCCAA-3′) and 706R (5′-AATCCRAGAATTTCACCTCT-3′) primers. The purified PCR products were sent to Novogene Co., Ltd. (Beijing, China) for Illumina HiSeq2500 platform sequencing (Illumina, San Diego, CA, USA). All samples were repeated in triplicate.

The effective data were denoised using the Divisive Amplicon Denoising Algorithm 2 (DADA2), and the final amplicon sequence variants (ASVs) were then generated by removing the redundant and low occurrence (*n* < 5 within all samples). QIIME2’s class-sklearn algorithm was used for species annotation for each ASV using a pre-trained Naive Bayes classifier.

### 2.3. Rice Wine Fermentation on a Lab-Scale

The main brewing process of turbid rice wine is depicted in Figure 2. At room temperature, 250 g glutinous rice was soaked in distilled water for 4 h, then drained and steamed for 35 min at 100 °C. After the glutinous rice cooled down to room temperature, it was mixed with 1 g Qu and transferred into a pottery jar. Then, the mixture was supplemented with 200 mL distilled water to ferment at 25 °C for 3 days. A 50 mL fermentation broth of each TRW sample was stored at 4 °C prior to testing. TRW fermented by GQ, DQ, AQ, and LQ was named GW, DW, AW, and LW, respectively. All experiments were repeated in triplicate.

### 2.4. Reducing Sugar and Ethanol Analysis

The reducing sugar was determined by the 3,5-dinitrosalicylic acid (DNS) assay [5]. The ethanol content was determined using a gas chromatograph (GC-2020 Plus, Shimadzu, Japan) equipped with a capillary column of WONDA CAP WAX (30 m × 0.25 mm × 0.25 µm) and detector of DET1 [9].

### 2.5. Non-Volatile Organic Acids Analysis

Organic acid analysis was carried out via HPLC (Waters 2695, Waters Corporation, Milford, MA, USA) equipped with an Aminex HPX-87H ion exchange column (300 × 7.80 mm, 9 μm film thickness, Bio-Rad Laboratories, Inc., Hercules, CA, USA) at a temperature of 60 °C [10]. The injection volume was 20 μL, and the mobile phase was 7 mmol/L H_2_SO_4_ solution (pH 2.2) at the flow rate of 0.60 mL/min. The ultraviolet detector was set at 210 nm.

### 2.6. Volatile Compounds Analysis

The volatile compounds were extracted via headspace solid phase microextraction (SPME) equipped with a 75 μm Carboxen/PDMS StableFlex fiber (Supelco, Bellefonte, PA, USA) for 30 min at 80 °C and transferred to a gas chromatography inlet to desorb at 250 °C for 10 min. Then, they were analyzed by an Agilent 6890N GC coupled with an Agilent 5973i quadrupole mass detector (Agilent Technologies, Inc, Palo Alto, CA, USA) [10]. The separations were carried out by a HP-5MS capillary column (30 m × 0.25 mm, 0.25 μm film thickness, Agilent Technologies, Inc, Palo Alto, CA, USA).

### 2.7. Statistical Analyses

The differential fungi and volatile compounds in samples were discovered via partial least squares discriminant analysis (PLS-DA) with SIMCA software (version 14.1) (Umetrics, MKS Umetrics AB, Umea, Sweden), then the important volatile compounds with variable importance for the projection (VIP) > 1.00 were visualized by circus (http://mkweb.bcgsc.ca/tableviewer/visualize/ accessed on 2 August 2022). The fungal biomarkers in Qu were analyzed by linear discriminant analysis effect size (LEfSe) with the Kruskal–Wallis test (*p* < 0.05), Wilcoxon test (*p* < 0.05), and LDA threshold score > 2.00 on the Galaxy website (http://huttenhower.sph.harvard.edu/galaxy/, accessed on 2 August 2022). The chemical data were analyzed using IBM SPSS software (version 21) by one-way analysis of variance (ANOVA) with least significant difference (LSD) test at *p* = 0.05. Correlation analysis was performed using Spearman’s correlation between fungal communities and volatile compounds on the online platform of OmicShare tools (https://www.omicshare.com/tools/Home/Soft/ica2, accesses on 3 September 2022) and visualized via the Cytoscape software (version 3.9.1).

## 3. Results

### 3.1. Fungal Profiles of Different Qu

A total of 25 fungi genera and 28 fungal species were found in different Qu from the rim of the Sichuan Basin (Appendix A and Figure 3A). The dominant genera were *Rhizopus*, *Lichtheimia*, *Saccharomycopsis*, *Saccharomyces*, and *Candida*, which accounted for more than 96% of the total abundance of fungi in each Qu. At species level, *Rhizopus arrhizus* (*R. arrhizus*) was predominant in Qu, accounting for 75.1%, 65.0%, 49.8%, and 53.8% in GQ, DQ, AQ, and LQ, respectively. However, the sub-dominant fungi in Qu were obviously different, which were *Candida glabrata* (*C. glabrata*, 10.4%) and *Saccharomycopsis fibuligera* (*S. fibuligera*, 7.47%) in GQ, *S. fibuligera* (27.9%) and *Rhizopus microsporus* (*R. microspores*, 3.84%) in DQ, *Lichtheimia ramose* (*L. ramose*, 41.8%) and *Saccharomyces cerevisiae* (*S. cerevisiae*, 6.24%) in AQ, and *R. microsporus* (21.6%) and *S. cerevisiae* (20.0%) in LQ (Figure 3A). These results indicated that the fungal structure in Qu was significantly affected by the different geographical environments around the Sichuan Basin. The PLS-DA further showed that fungal communities were obviously divided into four groups based on all fungi in Qu, where R^2^X and R^2^Y were 0.664 and 0.966, respectively, and Q^2^ of the model was 0.911 (Figure 3B). Among these species, 12 fungi were verified as biomarkers (Figure 3C) and most dominant species belonged to biomarkers. *R. arrhizus* (75.1%), *C. glabrata* (10.4%), *Rhizomucor pusillus* (*R. pusillus*, 0.469%), *Pichia kudriavzevii* (*P*. *kudriavzevii*, 0.192%), *Lichtheimia corymbifera* (*L. corymbifera*, 0.257%), *Thermomyces lanuginosus* (*T. lanuginosus*, 0.456%), and *Wallemia sebi* (*W. sebi*, 0.309%) were the biomarkers in GQ, *S. fibuligera* (27.9%) and *Mucor indicus* (*M. indicus*, 1.18%) in DQ, *R. microspores* (21.6%) and *S. cerevisiae* (20.0%) in LQ, while AQ only had one biomarker of *L. ramose* (41.8%) (Figure 3C). In general, the predominant fungi and biomarkers in Qu might contribute to the different chemical characteristics in corresponding TRW.

### 3.2. The Chemical Indexes in TRW

Chemical indexes of rice wine often reflect the quality or type of TRW. Reducing sugars in GW and DW from the parallel fold ridge valley belt area (181.9–248.9 g/L) were obviously higher than AW and LW from the plateau area (121.6–128.1 g/L) (Figure 4A). By contrast, the ethanol contents in TRW fermented by Qu from plateau (11.4–12.4%) were higher than that from the parallel fold ridge valley belt area (4.56–5.85%) (Figure 4B). The composition and amounts of organic acids are also important contributors to the sour taste of rice wine [4,11]. A total of 7 organic acids were identified and quantified, including L-lactic acid, succinic acid, quinic acid, pyruvic acid, citric acid, tartaric acid and shikimic acid (Figure 4C). GW had the highest content of total organic acids (1.14 g/L), then followed by AW (0.921 g/L) and by DW (0.669 g/L) and then LW (0.553 g/L). L-lactic acid was absolutely predominant in GW (85.8%), DW (69.3%), AW (67.5%), and LW (41.2%). Except for GW (1.65%), succinic acid was also dominant in DW (19.0%), AW (22.8%), and LW (40.5%). Additionally, tartaric acid was only detected in GW and DW, but shikimic acid in GW.

### 3.3. Volatile Compound Profiles in TRW

Volatile compound, the main source of aroma, is one of the main factors affecting the quality characteristics of alcoholic products [12]. A total of 162 volatile compounds were identified in TRWs, including 27 alcohols, 48 esters, 10 aldehydes, 21 ketones, 17 acids, 17 alkanes, and 22 other compounds (Appendix A). Of these, 108 were found in GW, 92 in DW, 87 in AW, and 52 in LW (Appendix A). Except for 31 volatile compounds which were commonly shared in all TRWs, 34, 21, and 20 unique volatile compounds were discovered in GW, DW, and AW, respectively (Appendix A). Interestingly, there were not any unique compounds in LW. The AW had the highest content of total volatile compounds with 51.7 mg/L, followed by DW with 47.9 mg/L, LW with 37.2 mg/L, and GW with 20.5 mg/L (Figure 4D). Among them, alcohols contributed the greatest to DW (57.8%), AW (77.3%), and LW (58.2%), while there was only 25.2% in GW, in which acids were the most abundant, at 35.6%. In DW, AW, and LW, acids made up 2.45–6.68%. Esters were the second abundant volatile compounds, accounting for 19.1%, 13.2%, 10.3%, and 23.3% in GW, DW, AW, and LW, respectively. The ketones were higher in GW (10.6%) and DW (14.8%) than in AW (3.34%) and LW (6.43%). The aldehydes, alkanes, and other compounds made up 0.597–6.53%. Obviously, the different Qu led to different profiles of volatile compounds in TRWs.

All TRWs indicated distinct differences based on the types and contents of volatile compounds (Figure 5A). The 58 most important volatile compounds with VIP > 1.00 accounted for 79.0%, 82.1%, 92.5% and 88.8% in GW, DW, AW, and LW, respectively, including 12 alcohols, 19 esters, 5 aldehydes, 5 ketones, 8 acids, 3 alkanes, and 6 other compounds (Figure 5B,C and Figure 6). Among them, acetic acid (29.4%) and phenethyl alcohol (19.6%) were dominant in GW. However, they changed to phenethyl alcohol (29.3%), 3-methyl-1-butanol (20.6%), acetoin (7.70%), and (R, R)-2, 3-butanediol (6.20%) in DW, 3-methyl-1-butanol (56.5%) and phenethyl alcohol (14.1%) in AW, and 3-methyl-1-butanol (34.0%), phenethyl alcohol (17.0%), and ethyl palmitate (10.1%) in LW. The acetoin was 3.69 mg/L in DW; interestingly, it was not detected in other TRWs. Furthermore, the GW, DW, and AW had 4, 2, and 13 unique important volatile compounds (VIP > 1.00), respectively (Figure 6). Therefore, different geographical Qu also greatly affected the compositions of important volatile compounds in TRWs.

### 3.4. Correlations between Fungi and Volatile Compounds

Fungi in Qu are closely linked to the formation of volatile compounds in TRWs [7]. A total of 22 fungal species were significantly correlated with 158 volatile compounds (|r| ≥ 0.7, *p* < 0.05) (Figure 7A). Most biomarkers in Qu had positive correlation with almost all unique compounds in its corresponding TRW, which confirmed the contribution of biomarkers to the formation of specific flavor compounds.

Furthermore, 58 important volatile compounds exhibited correlations (|r| ≥ 0.7, *p* < 0.05) with the fungi present in Qu (Appendix A), which were visualized in Figure 7B. In GW, acetic acid, 2-methyl-2-phenyl-1,3-dioxolane, lauric acid, and 1-octanol were only positively correlated with one or more of the GQ biomarkers, while 2 (5H)-furanone, 2, 5-dimethylbenzaldehyde, formic acid, methyl 2-ethylacetoacetate, methyl pyruvate, and 1-methoxy-2-propanol were positively correlated with the GQ biomarkers, but also with one or more of non-biomarkers, including *Aspergillus penicillioides*, *Rhizomucor miehei*, *Diutina rugosa*, *Millerozyma farinose*, and *Syncephalastrum monosporum*. In DW, acetoin and ethyl 5-chloro-1,3,4-thiadiazole-2-carboxylate were positively associated with DQ biomarkers (*S. fibuligera* and *M. indicus*) and the non-biomarker *Saccharomycopsis malanga*, but negatively correlated with the GQ biomarker *R. pusillus*, *T. lanuginosus*, and the AQ biomarker *L. ramosa*. In AW, stearic acid, propylene glycol, and all unique important volatile compounds of AW were only positively correlated with the AQ biomarker *L. ramose*, and all unique important volatile compounds of AW had negative correlations with DQ biomarkers. 3-Methyl-1-butanol, ethyl oleate, (S)-(+)-citramalic acid, (S)-(+)-1,2-propanediol, 4-hydroxyphenethyl alcohol, ethyl myristate, ethyl acetate, furaneol, and farnesol were only positively correlated with LQ biomarker *S. cerevisiae*; cis-1, 2-cyclohexanediol, methyl isobutyrate, methyl acrylate, and S-methyl thioacetate had positive correlations with the LQ biomarker *R. microspores* and the non-biomarker *Aspergillus niger* (*A. niger*), while ethyl palmitate, ethyl linoleate, and benzaldehyde were all positively correlated with *S. cerevisiae*, *R. microspores*, and *A. niger*.

## 4. Discussion

Fungi in Qu are considered a crucial influence on the quality of TRW [5,6,13]. As one of the main TRW production regions in China, the Sichuan Basin and its surrounding areas has formed its own characteristics of TRWs; unfortunately, the fungi in Qu and how they affect the specific compound formation in TRWs had never been elucidated. Therefore, analysis of fungi in Qu and further exploration of their correlations with chemical characteristics of TRW are essential to deeply understand the cause of the regional TRW formation around the Sichuan Basin.

The fungal community of Qu was distinctly different in various geographical locations [2,6,14,15,16]. Thus, fungal biomarkers could distinguish different Qu [6,17]. In this study, fungal biomarkers in Qu were obviously different (Figure 3C). GQ had the most biomarkers, which might be the reason for the most unique volatile compounds in GW (Appendix A). Additionally, *C. glabrata* and *S. fibuligera*, as the biomarkers of Qu from hill and mountain areas, were also widely found in Hong Qu from hill areas in Fujian province [18,19,20]. *L. ramose*, *S. cerevisiae*, and *R. microspores*, as biomarkers of Qu from plateaus, were commonly found in Da Qu around China [21]. In general, the fungal communities in Qu from the rim of the Sichuan Basin varied between geographic locations.

Responding to the differences in fungal communities, the chemical characteristics of TRWs had distinct difference. *R. arrhizus* can produce L-lactic acid [22]. It was the most abundant fungus in GQ, which might result in the large amount of L-lactic acid in GW (Figure 4C). *S. fibuligera* can secrete amounts of extracellular hydrolases to hydrolyze polysaccharides into sugars [23,24], and then sugars are converted into ethanol by *S. cerevisiae* [25]. In GQ and DQ, *S. fibuligera* was in higher relative abundance, while *S. cerevisiae* was in low abundance (Figure 3A), resulting in higher reducing sugars and lower ethanol content in GW and DW (Figure 4A,B). *C. glabrata* could utilize both glucose and xylose to produce alcohols [2,26]. It had higher abundance than *S. cerevisiae* in GQ, which might be the main producer of alcohols in GW. *L. ramose* and *R. microsporus* can produce hydrolases to hydrolyze polysaccharides into fermentable sugars [27,28,29], which would then be efficiently converted to ethanol by *S. cerevisiae*. The high abundance *L. ramose* and *R. microsporus* in AQ and in LQ, respectively, together with the higher abundance *S. cerevisiae*, resulted in higher ethanol content and lower reducing sugars in AW and LW (Figure 3A and Figure 4A,B). Obviously, those predominant fungi in Qu led to different chemical characteristics of TRWs.

Volatile compounds were one of the main influences on the sensory characteristic of TRW [1]. Of the important volatile compounds (VIP > 1.00), phenethyl alcohol, was dominant in all TRWs (14.1–29.3%) (Figure 6), which might play a vital role in the flavor of TRW, since it had quietly elegant rosy and honey aromas and was used as an important characteristic component of rice wine [5]. 3-Methyl-1-butanol, one of the major aliphatic alcohols [13] with a banana flavor [30], had been characterized as an important aroma compound in Qingke liquors from the Qinghai–Tibetan plateau [31], which also showed high proportions in TRWs except GW. It accounted for the highest proportion in AW (56.5%) and LW (36.0%) (Figure 6), suggesting that it might be the most important compound affecting the specific flavor of AW and LW from plateaus. However, in GW, the acetic acid with vinegar aroma was the highest, which might lead to the different flavor of GW from other TRWs with lower acetic acid content (Figure 6). Additionally, unique important compounds in TRWs might be one of the reasons for the characteristic flavor of TRWs. For instance, formic acid in GW could endow it a sour, strong irritation flavor [32], and acetoin in DW could give it a pleasant buttery odor [33]. In LW, although there were no unique compounds, the esters such as ethyl palmitate, ethyl oleate, and ethyl linoleate with higher content might give a unique flavor to LW (Figure 6). These esters have their own aromas, for example, ethyl palmitate has fruity, candy, and perfume-like aromas [8]; ethyl oleate has a faint, floral note; ethyl linoleate has a waxy, creamy, fatty, coconut odor [34]. Therefore, the dominant or unique important volatile compounds, which made important contributions to the distinctive flavor of TRWs, were a significant difference in TRWs fermented by different regional Qu.

The metabolism of fungi in Qu can produce complex compounds during TRW brewing, and a compound is usually the result of combined actions of fungi [8]. Of GQ biomarkers and important volatile component variable (VIP > 1.00), the acetic acid with the highest content in GW (6.04 mg/L) (Figure 6) was positively correlated with *R. arrhizus*, *R. pusillus*, *T. lanuginosus*, and *P. kudriavzevii* (Figure 7B, Appendix A). Additionally, the unique important compounds of GW, namely formic acid, methyl 2-ethylacetoacetate, methyl pyruvate, and 1-methoxy-2-propanol, all had highly positive correlations with the GQ biomarkers *R. arrhizus*, *C. glabrata*, *R. pusillus*, *T. lanuginosus*, and *W. sebi* (Figure 7B, Appendix A), which directly certified the functions of these fungi on the formation of unique flavor in GW. Among those biomarkers, *R. arrhizus*, as the most abundant fungus in GQ, could hydrolyze racemic acetates to produce (R)-(+)-alcohols and acetic acid [35], which might be the main cause of the massive amount of acetic acid in GW (Figure 3A, Figure 6 and Appendix A). *C. glabrata*, the second dominant fungus in GQ, could produce more esters and important terpene substances with the high activity of β-glucosidase [36], which might increase the floral and fruity aroma in GW. In DW, the unique important compounds acetoin and ethyl 5-chloro-1,3,4-thiadiazole-2-carboxylate were positively associated with the DQ biomarkers *S. fibuligera* and *M. indicus* (Figure 7B, Appendix A). Acetoin, in particular, with a pleasant buttery odor and as a key aroma contributor in wines [33,37] showed the high ratio of 7.70% in DW (Figure 6), which implied its key contribution to the special flavor of DW. The content of acetoin in Zaopei increased when *S. fibuligera* was inoculated in Sichuan-style Xiaoqu [23], suggesting that *S. fibuligera* with high abundance in DQ might also produce acetoin in DW by itself or by influencing other microorganisms. In AW, all unique important volatile compounds were positively correlated with *L. ramose* (Figure 7B, Appendix A). *L. ramose*, as one of the main functional microbes involved in the main flavor compounds’ development in Daqu [38], was also predominant in AQ and was determined as a biomarker (Figure 3), which implied it was irreplaceable in generating the unique flavor of AW. In LW, the ethyl oleate, (S)-(+)-citramalic acid, and furaneol in higher proportions had positive correlations with *S. cerevisiae*; methyl isobutyrate and methyl acrylate in higher proportions were positively correlated with *R. microspores*; and ethyl palmitate (10.1%) and ethyl linoleate (3.35%) had positive correlations with *S. cerevisiae* and *R. microspores* (Figure 6 and Figure 7B, Appendix A). *S. cerevisiae*, as an effective ethanol producer, is widely used in making wine, bread, and beer [17] and could produce secondary metabolites, such as amino acids, organic acids, and volatile flavor substances [39]. *R. microsporus* could secrete amylases to hydrolyze starch into sugar for further microbial utilization [29] and was positively correlated with volatile alcohols, acids and esters [18]. Moreover, it could produce lipase to synthesis ethyl oleate under solid-state fermentation [40], which might be the reason for the higher content of ethyl oleate in LW (Appendix A and Figure 6). Consequently, the LW biomarkers *S. cerevisiae* and *R. microspores* were the main contributory factors to the particular flavor production in LW. In short, most biomarkers in Qu showed strongly positive correlation to the important volatile compounds in corresponding TRW, thus proving their key role in the formation of the unique style of TRW.

Notably, the correlation analysis was based on statistical methods but did not imply that these compounds were directly produced by these fungal biomarkers, which might account for the positive association of some non-biomarkers with important volatile compounds (Appendix A). Metagenomics, transcriptomics, metabonomics, and community reconstruction can be used to further investigate the functions of these fungi.

## 5. Conclusions

In the current study, *R. arrhizus* was found absolutely predominant in Qu from different regions in the rim of the Sichuan Basin; however, Qu from different areas had different fungal biomarkers. The biomarkers in GQ mainly were *R. arrhizus*, *C. glabrata*, *R. pusillus*, *T. lanuginosus*, and *W. sebi*, while *S. fibuligera* and *M. indicus* were the biomarkers in DQ, *L. ramose* in AQ, and *R. microsporus* and *S. cerevisiae* in LQ. Responding to fungal biomarkers, the chemical characteristics were also markedly changed in TRWs. GW and DW had higher reducing sugars and lower ethanol contents, which were opposite to AW and LW. Among important volatile compounds (VIP > 1.00), acetic acid (29.4%) and phenethyl alcohol (19.6%) were dominant in GW. However, they shifted to phenethyl alcohol (29.3%), 3-methyl-1-butanol (20.6%), acetoin (7.70%), and (R, R)-2, 3-butanediol (6.20%) in DW, 3-methyl-1-butanol (56.5%) and phenethyl alcohol (14.1%) in AW, and 3-methyl-1-butanol (34.0%), phenethyl alcohol (17.0%), and ethyl palmitate (10.1%) in LW. Moreover, biomarkers in Qu showed strongly positive correlation to the important volatile compounds in corresponding TRW. Among these, GQ and AQ biomarkers were positively correlated with all unique important volatile compounds in GW and AW, respectively. Meanwhile, GQ biomarkers had positive correlation with acetic acid, DQ biomarkers with acetoin and ethyl 5-chloro-1,3,4-thiadiazole-2-carboxylate, and LQ biomarkers with 16 important volatile compounds in LW. This study provides a more comprehensive and in-depth insight into the regional Qu from the rim of the Sichuan Basin and may improve the quality and flavor of TRW by regulating the key fungi in Qu.

## Figures and Tables

**Figure 1 foods-12-00585-f001:**
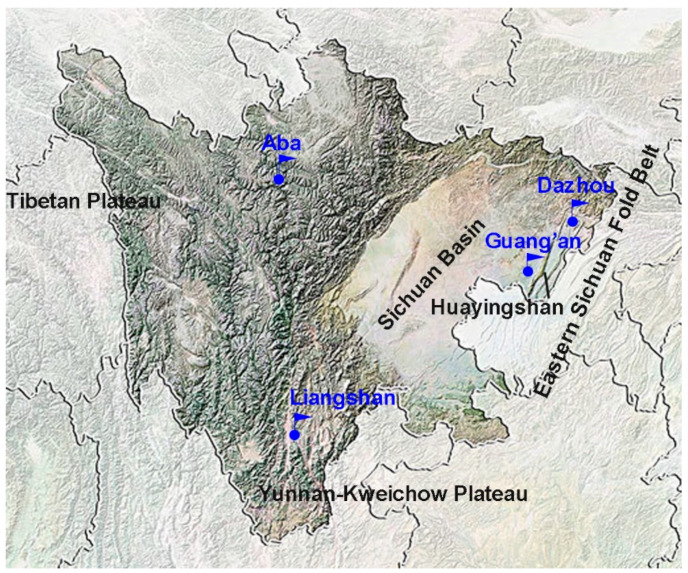
Sampling distribution of the four Qu from different geographical areas in the rim of the Sichuan Basin.

**Figure 2 foods-12-00585-f002:**
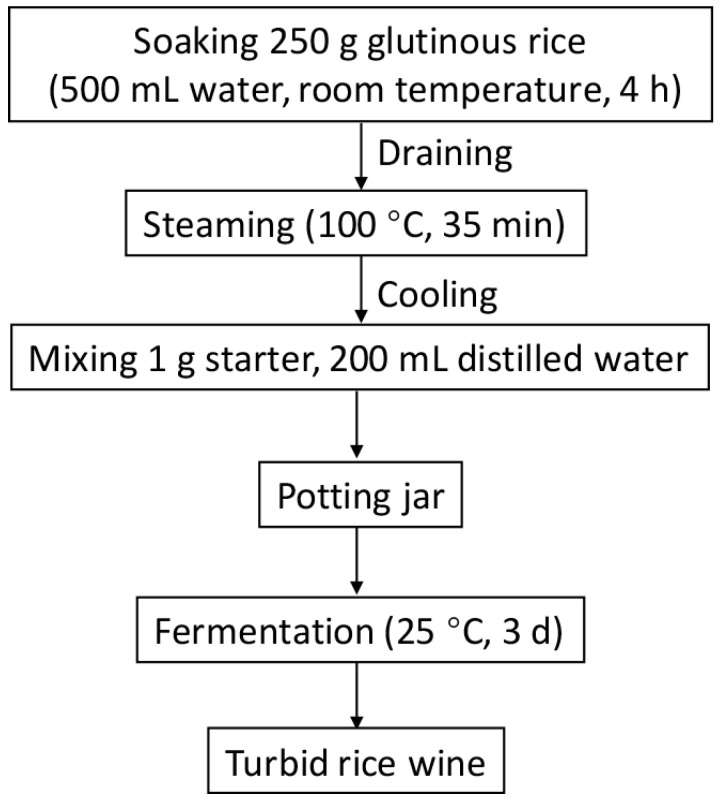
Schematic diagram of turbid rice wine fermentation.

**Figure 3 foods-12-00585-f003:**
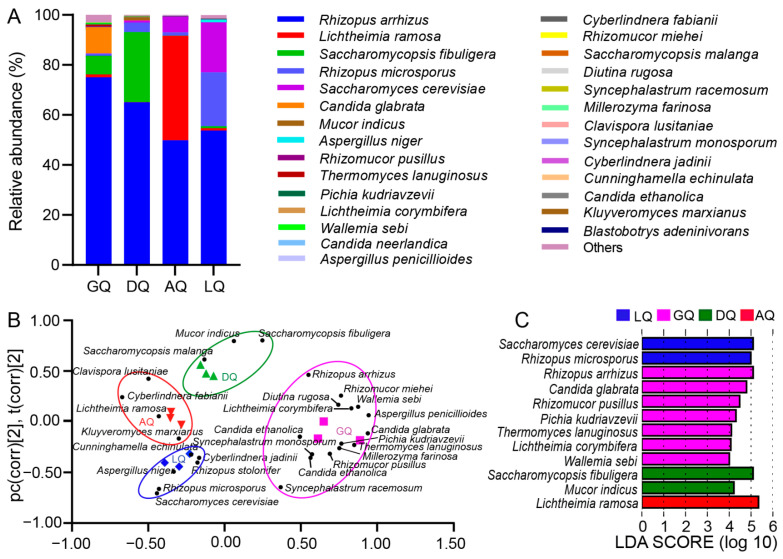
Analysis of fungal species composition in different geographical Qu. (**A**) Relative abundance of fungal species identified in different Qu. (**B**) Partial least squares discriminant analysis (PLS-DA) score plot of fungi among different Qu. (**C**) The fungal biomarkers analyzed using the linear discriminant analysis (LDA) effect size (LEfSe) method by the statistically significant LDA threshold of >2.0. Results are shown as the mean from three biological replicates.

**Figure 4 foods-12-00585-f004:**
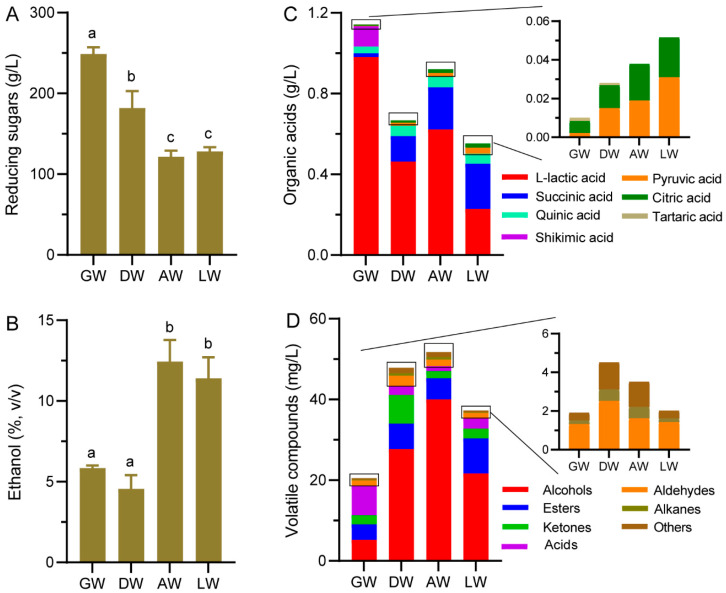
The chemical indexes of rice wine fermented by different Qu. (**A**) Reducing sugars, (**B**) ethanol, (**C**) organic acids, (**D**) volatile compounds. Bars with different superscript letters within a figure indicate significantly different (*p* < 0.05). Results are shown as the mean from three biological replicates.

**Figure 5 foods-12-00585-f005:**
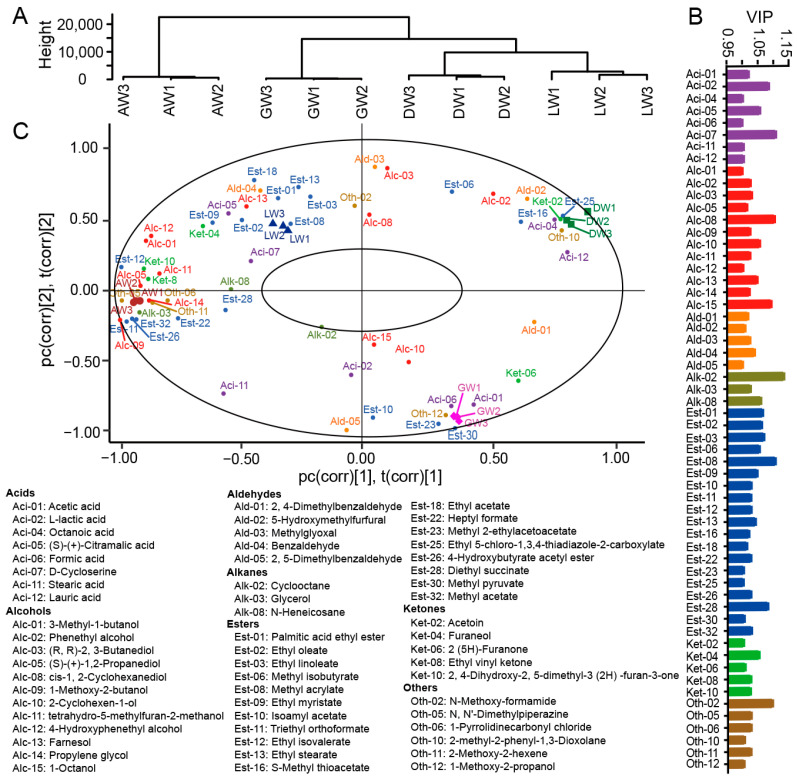
Analysis of volatile compounds in rice wine samples fermented by different regional Qus. (**A**) Hierarchical clustering of volatile compounds was performed using the method of average linkage; (**B**) the variables important in the projection (VIP) values of the 58 important volatile compounds (VIP > 1.00); (**C**) biplot of volatile compounds with VIP value > 1.00 from partial least squares discriminant analysis (PLS-DA) model among different samples.

**Figure 6 foods-12-00585-f006:**
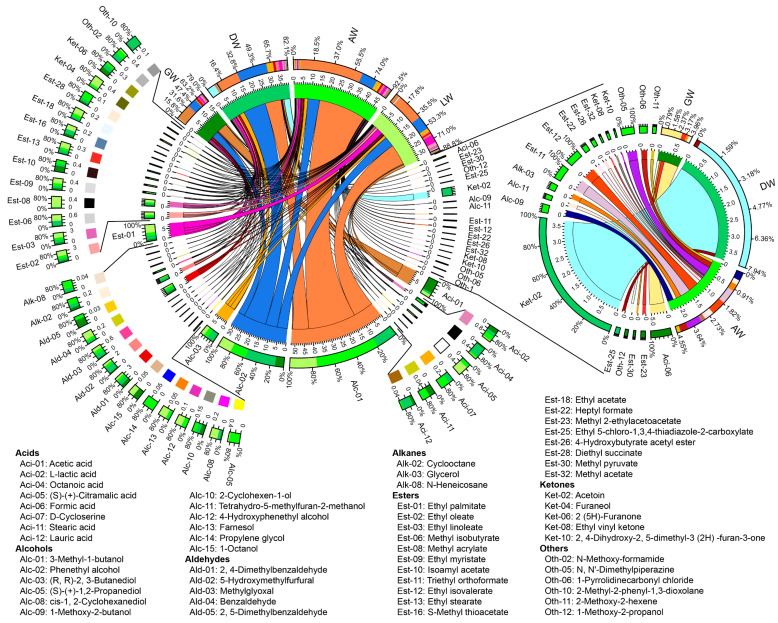
Distribution of the 58 important volatile compounds for each rice wine sample. The width of the bars from each compound indicates the distribution ratio of that compound in the sample. The unit of the inner ring is mg/L.

**Figure 7 foods-12-00585-f007:**
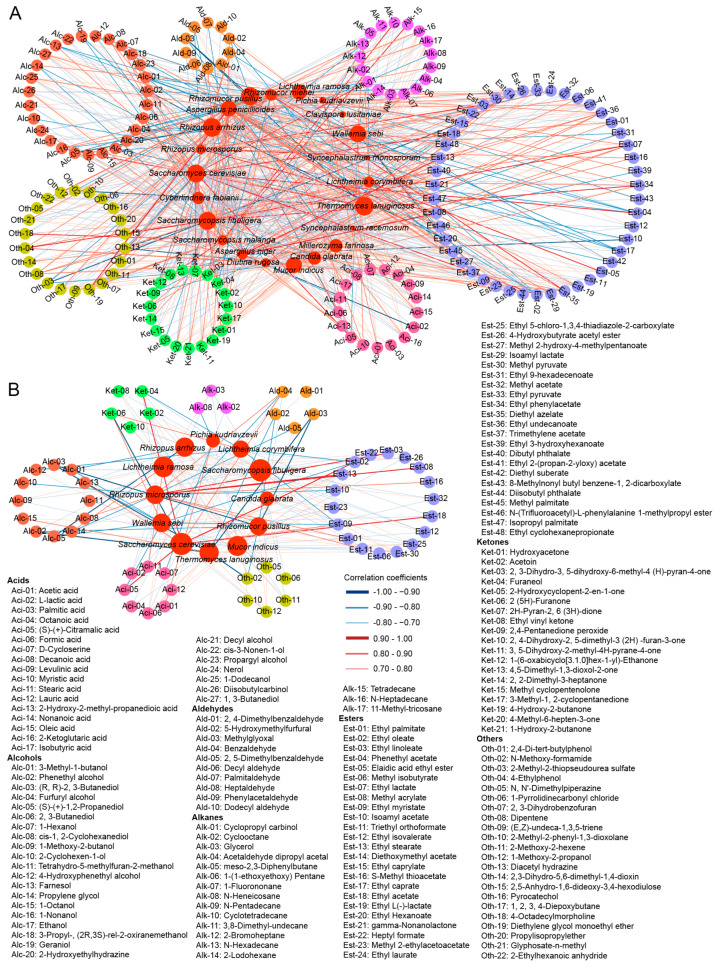
Visualization of the correlation network according to significant correlations between fungal species and volatile compounds in corresponding rice wine samples. (**A**) All fungal species and volatile compounds; (**B**) fungal biomarkers and important volatile compounds (VIP > 1.00). Only significant edges are drawn in the network using the Spearman’s correlation test (|r| ≥ 0.7 with *p* < 0.05). The size of the nodes representing fungi (red circle) indicates the size of degree value.

## Data Availability

Raw data obtained in this study are available from the corresponding author upon reasonable request.

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
