# Peer review of "Fungal Biomarkers in Traditional Starter Determine the Chemical Characteristics of Turbid Rice Wine from the Rim of the Sichuan Basin, China"

_foods, 2023, doi:10.3390/foods12030585_

Round 1

Reviewer 1 Report

The submitted paper explores the microbial communities of Chinese turbid rice wine from different geographical locations. The investigation of traditionally fermented products is an area of considerable interest now and this study will add useful knowledge to this topic. The methodological approach has generated a significant amount of data which are not always presented in the clearest manner (for example Figure 2), but I am currently unaware of more appropriate presentation formats.

There are several questions to be answered or suggestions that I believe may improve the submission prior to publication:

The title suggests that the study was looking at physicochemical characteristics of TRW, but the analysis seems to be simply chemical. Which analysis pertains to the physical properties of the drink as opposed to the chemical composition?

The descriptor ‘biomarkers’ is used repeatedly through the manuscript. One definition of a biomarker is a measurable indicator of some biological state or condition. This paper uses the term to refer to the microbiological composition of the culture in TRW. I’d suggest that the cultures may produce biomarkers, such as the flavour compounds discussed, but they themselves are better referred to is microorganisms present, microbial load/composition or similar.  

A key concluding remark is that “This study provided a rationale for promoting the excellent characteristics of TRW by controlling beneficial fungal communities.”. It is not immediately clear from the paper if this statement is justified. Has consumer preference been measured, or appropriate references identified which have looked at this aspect? If it is not clear what needs to be optimised for then knowledge of microbial composition and resulting fermentation characteristics is unlikely to help.

Revision of several individual points should be considered (listed below):

Line 107: a picture of the fermentation set-up may provide clarification.

Line 212: Sub-abundant- not clear what is meant by this and it is not a commonly used phrase.

Line 225: The 58 MOST important…

Line 292: Consider revising for clarity.

Line 342: sub-dominant- not clear what is meant by this and it is not a commonly used phrase.

Line: 353: “…the main flavour development…” No sensory analysis was completed therefore overall flavour as perceived by humans was not assessed. Presumably this should be “the main flavour compound development…”

Line 373: un-biomarkers- not clear what is meant by this and it is not a commonly used phrase.

Line 358: were positively correlated with…

Reviewer 2 Report

What is the novelty and originality of this work? Which should be clarified in the introduction

The abstract section lacks background and a gap

The authors must expound on their findings and results.

No reference from the Foods journal was added, therefore it does not present relevance with this journal

The conclusion section is insufficient

Therefore, I cannot recommend the submitted manuscript is published in Foods in this way.

Reviewer 3 Report

The paper is easily readable in style, language and presentation. The overall publication shows reasonable experimental design and results. 

The experimental part is clearly structured and makes sense.The data is presented in a comprehensive way and is explained in high detail. 

Quantity and quality of reference und literature is given in an adequate way.

The only comment is In paragraph 2.3 please mention if the fermentation was aerobic (and how you controlled the oxygen in that case) or not aerobic. 

Reviewer 4 Report

The manuscript is very well written and offer a lot of scientific results regarding the production of TRW in Sichuan Basin, China.

I have few questions regarding the Rice wine fermentation on a lab-scale.

1. How you obtain the Qu starter culture, please provide more information regarding traditional process.

2.What is the reason that you used only 1 g of Qu during lab fermentation process. Why you haven't variants with other quantity of Qu.

3.Fermentation temperature for TRW is 25 0C, please explain the value. The Qu is a mixt between yeasts and molds. In general the yeast prefer for fermentation temperature between 20-22 0C for white wine technology and 250C for red wine technology. The molds can activate at different temperature from 4-380C. Why you used exact 25 0C for fermentation process.

I recommend minor revision.
